# Temozolomide and Lomustine Induce Tissue Factor Expression and Procoagulant Activity in Glioblastoma Cells In Vitro

**DOI:** 10.3390/cancers15082347

**Published:** 2023-04-18

**Authors:** Maaike Y. Kapteijn, Shanna Zwaan, Esther ter Linden, El Houari Laghmani, Rob F. P. van den Akker, Araci M. R. Rondon, Sabina Y. van der Zanden, Jacques Neefjes, Henri H. Versteeg, Jeroen T. Buijs

**Affiliations:** 1Einthoven Laboratory for Vascular and Regenerative Medicine, Division of Thrombosis & Hemostasis, Department of Internal Medicine, Leiden University Medical Center, 2333 ZA Leiden, The Netherlands; m.y.kapteijn@lumc.nl (M.Y.K.);; 2Department of Cell and Chemical Biology, ONCODE Institute, Leiden University Medical Center, 2333 ZC Leiden, The Netherlands

**Keywords:** cancer-associated thrombosis, venous thromboembolism, glioblastoma, chemotherapy, tissue factor, extracellular vesicles, temozolomide, lomustine

## Abstract

**Simple Summary:**

Cancer patients, particularly with glioblastoma (GBM), are at increased risk for thrombosis. This risk is further increased upon treatment with chemotherapy. Tissue factor (TF) is the initiator of the extrinsic coagulation pathway and expressed by GBM cells. Chemotherapy is known to affect TF expression and activity in several cancer types, but the effect of the commonly used chemotherapeutic agents Temozolomide and Lomustine on TF in GBM is unknown. In this study, we describe a promoting role for these agents on TF procoagulant activity and TF gene and protein expression in three GBM cell lines. This may explain the increased risk of thrombosis in GBM patients undergoing chemotherapy. Our findings may have implications for future GBM treatment since patients with high TF levels may benefit from anticoagulant treatment by decreasing the risk of chemotherapy-induced thrombosis in GBM.

**Abstract:**

Glioblastoma (GBM) patients have one of the highest risks of venous thromboembolism (VTE), which is even further increased upon treatment with chemotherapy. Tissue factor (TF) is the initiator of the extrinsic coagulation pathway and expressed by GBM cells. In this study, we aimed to examine the effect of routinely used chemotherapeutic agents Temozolomide (TMZ) and Lomustine (LOM) on TF procoagulant activity and expression in GBM cells in vitro. Three human GBM cell lines (U-251, U-87, U-118) were exposed to 100 µM TMZ or 30 µM LOM for 72 h. TF procoagulant activity was assessed via an FXa generation assay and TF gene and protein expression through qPCR and Western blotting. The externalization of phosphatidylserine (PS) was studied using Annexin V flow cytometry. Treatment with TMZ and LOM resulted in increased procoagulant activity in all cell lines. Furthermore, both agents induced procoagulant activity in the supernatant and tumor-cell-secreted extracellular vesicles. In line, TF gene and protein expression were increased upon TMZ and LOM treatment. Additionally, PS externalization and induction of inflammatory-associated genes were observed. Overall, the chemotherapeutic modalities TMZ and LOM induced procoagulant activity and increased TF gene and protein expression in all GBM cell lines tested, which may contribute to the increased VTE risk observed in GBM patients undergoing chemotherapy.

## 1. Introduction

Glioblastoma (GBM) is the most aggressive type of primary brain cancer, which coincides with a high risk of venous thromboembolism (VTE; 10–30%) [1,2]. The exact VTE risk depends on patient-related, tumor-related and therapy-related risk factors [3], including chemotherapy. Importantly, cancer-associated thrombosis (CAT) is a leading cause of death in patients with active cancer undergoing chemotherapy, after cancer progression itself [4]. Cancer patients receiving systemic chemotherapy during the first four months after diagnosis have an additional 3.4-fold increased risk of developing VTE [5], which may increase to a 4.5- to 6-fold enhanced risk depending on the exact treatment protocol [6]. Indeed, the most commonly used risk prediction model for cancer-related VTE, from Khorana et al., specifically focuses on cancer outpatients receiving chemotherapy [7], thus underlining the role of chemotherapy in CAT.

Chemotherapy increases the risk of VTE by damaging the vascular endothelium and inducing a pro-inflammatory state [8,9]. Tissue factor (TF), the initiator of the extrinsic coagulation cascade, is suggested to participate in chemotherapy-induced procoagulant activity observed in cancer patients [9,10]. Chemotherapy-mediated apoptosis leads to exposure of phosphatidylserine (PS) on the cell membrane, which is known to contribute to TF decryption and procoagulant activation [11,12,13]. Furthermore, in addition to apoptosis, cells may enter a state of replicative or accelerated senescence upon chemotherapeutic treatment [14]. An increased release of pro-inflammatory cytokines, such as TNF-α, IL-1β and VEGF, by tumor and endothelial cells indirectly also contributes to a prothrombotic state in chemotherapy-treated cancer patients [9,15]. Indeed, several chemotherapeutic agents, such as doxorubicin, gemcitabine, cisplatin and etoposide, have been linked to TF upregulation, increased TF activity and/or secretion of TF-positive extracellular vesicles (EVs) in a variety of cancer types in vitro [16,17,18,19]. In GBM, increased TF expression levels and TF-bearing EVs have also been observed [20], but, so far, a relation between chemotherapeutic treatment and TF upregulation has not been established.

In addition, brain cancer, and specifically GBM, is characterized by a unique pro-angiogenic and inflammatory microenvironment, which contributes to tumor progression, immune evasion and therapeutic resistance [21,22]. This, in combination with the high degree of genetic and epigenetic intratumoral heterogeneity, which is expressed in the different GBM genome-based subtypes (i.e., proneural, mesenchymal and classical) with unique immune compositions [23], results in a highly diverse GBM tumor for which successful treatment strategies are lacking. The hypercoagulable state, as imposed by TF, significantly exacerbates the disease burden, thus warranting more research on the potential role of chemotherapy-induced procoagulant activity and inflammation in GBM.

The most commonly used chemotherapeutics in GBM are temozolomide (TMZ) and lomustine (LOM). The standard of care for patients with primary GBM is TMZ with surgery and radiotherapy, whereas LOM is generally prescribed to patients suffering from recurrent GBM [24,25]. TMZ and LOM are both alkylating agents that induce DNA strand disassembly by methylating the purine bases of DNA, thereby preventing cell replication and ultimately leading to DNA double-strand breaks and cell death [26]. In contrast to TMZ, LOM is a nitrosourea compound that can create DNA inter-strand crosslinks and promote amino acid carbamoylation, thereby affecting key enzymatic functions within the cell [25].

Despite the general recognition of chemotherapy as an independent risk factor for CAT and the high risk of VTE in GBM, the influence of TMZ and LOM on the hypercoagulable state in GBM patients is unknown. Therefore, we investigated the effect of TMZ and LOM treatment on TF procoagulant activity and TF gene and protein expression in three GBM cell lines: U-251, U-87 and U-118. In addition, cellular processes in response to TMZ and LOM were studied since these might contribute to TF induction. We show that both chemotherapeutic modalities induce TF-mediated procoagulant activity and expression in all GBM cell lines tested, including PS externalization and upregulation of inflammatory-associated genes. These features may contribute to the increased risk of VTE in GBM patients undergoing chemotherapy.

## 2. Materials and Methods

### 2.1. Cell Culture

The human GBM cell lines U-87 and U-118 were derived from the American Tissue Culture Collection. The U-251 cell line was kindly provided by Prof. Dr. Janusz Rak (McGill University, Canada). All cell lines were maintained in DMEM (Dulbecco’s Modified Eagle Medium, 4.5 g/L D-glucose, L-glutamine, pyruvate; Thermo Fisher Scientific, Bleiswijk, The Netherlands), supplemented with 10% fetal calf serum (FCS; Bodinco, Alkmaar, The Netherlands), 1% penicillin–streptomycin (Gibco, Thermo Fisher Scientific) and 1% L-Glutamine (Merck, Darmstadt, Germany), in a humidified incubator (5% CO_2_) at 37 °C. Cell line identity was confirmed by Short Tandem Repeat (STR) profiling. Absence of mycoplasma was tested monthly. Cell numbers were adapted to surface area per experiment.

24 h after plating, cells were treated with the respective chemotherapeutics for 72 h, after which cells were processed for further experiments. In experiments with supernatant or EVs, medium with FCS was replaced by medium without FCS at 72 h after treatment. This was left on the cells for 2 h, followed by centrifugation at 1.000 g and 4 °C for 10 min to remove cell debris and further processing for subsequent read-out assays.

For EV-enriched samples, 1 × 10^6^ cells were plated in T175 cell culture flasks with 20 mL of supplemented DMEM. After removal of cell debris, ultracentrifugation at 20.000 g for 1 h at 4 °C was performed using 26.3 mL polycarbonate bottles (Beckman Coulter, Woerden, The Netherlands) in a 50.2 Ti rotor (without break) and an Optima XE-90 ultracentrifuge (Beckman Coulter).

Lomustine (LOM) was obtained from Santa Cruz (sc-202697) and Temozolomide (TMZ) from LUMC pharmacy. Both agents were dissolved in DMSO. Concentrations used (100 µM and 30 µM, respectively) were in the mid-range of LC50s, as described in the literature [27,28,29,30]. DMSO was used as vehicle (control).

### 2.2. FXa Generation

TF-dependent procoagulant activity was measured on attached cells, supernatant or isolated EVs. Cells and pelleted EVs were kept in HBS with 1.5 mM CaCl2. The assay was started upon addition of 1 nM FVIIa (Novo Nordisk, Alphen aan den Rijn, The Netherlands) and 50 nM FX (Stago, Leiden, The Netherlands). The reaction was quenched after 30 min at 37 °C, followed by measurement of the generated amount of FXa using the chromogenic substrate Spectrozyme^®^ FXa (Werfen, Breda, The Netherlands) and a kinetic recording spectrophotometer (Multiskan Ascent 354, Thermo Fisher Scientific). Measured rates were corrected for the number of cells present per sample. TF-blocking antibody mAb TF-5G9 was described previously [31]. Isotype-matched mouse IgG1 (TIB115) was used as control. Unlabeled human recombinant Annexin V was purchased from BioVision, Inc. (ITK diagnostics, Uithoorn, The Netherlands).

### 2.3. Quantitative Polymerase Chain Reaction (qPCR)

Total RNA of chemotherapy-treated cells was isolated using TRIsure (Bioline, GC biotech, Waddinxveen, The Netherlands) and the RNeasy Mini Kit (Qiagen, Venlo, The Netherlands) according to the manufacturer’s protocol. For cDNA synthesis, 1 µg RNA was converted using SuperScript II Reverse Transcriptase (Thermo Fisher Scientific). qPCRs were performed in a 384-well setup with SYBR Select Master Mix (Applied Biosystems, Bleiswijk, The Netherlands, Thermo Fisher Scientific) on a CFX384 Touch Real-Time PCR detection system (Bio-Rad, Lunteren, The Netherlands). Technical triplicates were included for every sample. See Appendix A for primer sequences used.

### 2.4. Western Blotting

Cells were lysed in Tris-Glycine SDS sample buffer 2X (Novex, Thermo Fisher Scientific). Lysates were denatured at 95 °C and sonicated for 10 s. Samples were loaded on BoltTM 4–12% Bis-Tris PLUS gels (Thermo Fisher Scientific), followed by transfer to 0.2 µm pore-size PVDF membranes (Bio-Rad). Membranes were blocked in TBS-T with 5% milk for at least 1 h at RT and incubated with the primary antibodies TF9-10H10, an anti-TF mouse monoclonal antibody (described in more detail in [31]), or the rabbit monoclonal antibody anti-GAPDH (14C10; Cell Signaling, Leiden, The Netherlands) in blocking buffer at 4 °C overnight. The next day, membranes were incubated with corresponding HRP-conjugated secondary antibodies (Abcam, Amsterdam, The Netherlands) for 1 h at RT. Bands were visualized with Western Lightning Plus-ECL (PerkinElmer, Groningen, The Netherlands) using a Bio-Rad ChemiDoc imaging system. ImageJ software (version 1.53q) was used for relative WB quantification.

### 2.5. Fluorescence-Activated Cell Sorting (FACS)

Chemotherapy-treated cells plated in T25 flasks were dissociated with TrypLE Express, with no phenol red (Gibco, Thermo Fisher Scientific). Cell suspensions were washed with Annexin V-binding buffer (10 mM HEPES, 140 mM NaCl and 2.5 mM CaCl2, pH 7.4) and counted. Per sample, 1 × 10^5^ cells were incubated with 0.25 µL PE-labeled Annexin V (BioLegend, Amsterdam, The Netherlands) in 50 µL binding buffer for 15 min at RT, followed by several washes with binding buffer. The percentage of Annexin V-positive cells was measured using a BD FACSCanto and analyzed using BD FACSDivaTM software.

### 2.6. Statistical Analysis

Statistical significance was measured using ordinary one- or two-way ANOVAs with Tukey’s multiple comparisons test. *p* < 0.05 was considered significant. GraphPad Prism version 9 was used for statistical evaluation.

## 3. Results

### 3.1. Chemotherapeutic Agents TMZ and LOM Induce Procoagulant Activity in GBM Cells

In order to determine the effect of chemotherapeutic agents TMZ and LOM on procoagulant activity in GBM cells in vitro, an FXa generation assay was performed using the human GBM cell lines U-251, U-87 and U-118. While basal levels of generated FXa were found to be higher in U-251 and U-118 cells compared to U-87 cells, treatment with TMZ and LOM for 72 h resulted in a significant increase in FXa generation levels in all three cell lines (*p* < 0.0001; Figure 1A).

Next to the observation that TMZ and LOM increased the procoagulant state of substrate-attached cells, we tested whether TMZ and LOM also affected the hypercoagulability of the supernatant. Hence, cells were treated with TMZ and LOM for 72 h, followed by a 2 h incubation with FCS-free medium to remove any procoagulant components in FCS. Subsequently, the supernatant was collected and centrifuged to remove cell debris. Treatment with both agents resulted in a significant increase in FXa generation in the supernatant of U-251 and U-118 cells as compared to the vehicle (*p* < 0.0001; Figure 1B). In the U-87 supernatant, FXa levels were again lower compared with the supernatant from U-251 and U-118 cells. In addition, only LOM treatment led to increased levels of FXa generation compared to the control (*p* < 0.0001).

Since it is observed that GBM cells can secrete TF-bearing EVs [20] and a link between EV secretion and certain chemotherapeutic modalities has been described [19], we wondered whether cancer cell-secreted EVs contributed to the observed increased procoagulant state of the supernatant. Therefore, EV-enriched samples were obtained by ultracentrifugation using the supernatant of TMZ- and LOM-treated cells. A significant increase in FXa generation was observed compared with EV samples isolated from vehicle-treated control cells from all cell lines and conditions (*p* < 0.0001 in EVs from all LOM-treated cells). In line with previous results, the effect of TMZ on EVs from U-87 cells (*p* < 0.05) was lower in comparison to U-251 and U-118 cells (*p* < 0.0001 in both cell lines; Figure 1C).

### 3.2. Treatment with TMZ or LOM Induces TF Gene and Protein Expression in GBM Cells

In addition to TF-mediated procoagulant activity, the effect of TMZ and LOM on TF gene and protein expression was determined using qPCR and Western blotting, respectively. A significant increase in TF mRNA expression levels was observed after exposure to both TMZ (U-251: *p* < 0.0001; U-87: *p* < 0.001; U-118: *p* < 0.0001) and LOM (U-251: *p* < 0.001; U-87: *p* < 0.0001; U-118: *p* < 0.0001; Figure 2A). Basal TF mRNA expression levels were lower in U-87 cells compared to the U-251 and U-118 cell lines.

In line with this, TF protein expression levels were also increased in TMZ- and LOM-treated GBM cells (Figure 2B,C and Appendix A for original WB membranes). A marked increase was observed with both agents in U-251 and U-118 cells (U-251: 3.1-fold with TMZ and 3.8-fold with LOM; U-118: 2.9-fold with TMZ and 3.2-fold with LOM; *p* < 0.0001 in all conditions). In U-87 cells, treatment with LOM resulted in a 1.2-fold increase in TF protein expression as compared with the vehicle (*p* < 0.0415).

### 3.3. Treatment with TMZ or LOM Increases the Percentage of Annexin V-Positive GBM Cells

To investigate whether chemotherapeutic treatment in GBM also affects secondary regulation of TF, the externalization of phosphatidylserine (PS) was determined as this is a known regulator of TF decryption [11,12]. Therefore, FACS analysis was performed on TMZ- and LOM-treated cells using PS-binding fluorescently labeled Annexin V (Figure 3). Both TMZ and LOM increased the percentage of Annexin V-positive cells in U-118 cells (*p* = 0.0023 and *p* = 0.0008, respectively). Comparable data were detected for U-251 cells following TMZ-treatment (*p* = 0.0044). No statistically significant differences were observed in TMZ- or LOM-treated U-87 cells.

### 3.4. TMZ- and LOM-Induced TF Activity Is Fully Prevented by TF Blocking Antibody mAb 5G9 and Reduced by PS-Binding Annexin V

To further untangle the chemotherapy-induced procoagulant response in GBM cell lines, we performed another FXa generation assay in the presence of the TF antagonistic antibody 5G9 (which inhibits TF-mediated coagulation), PS-binding Annexin V or TIB115 as IgG1 control (Figure 4). In the presence of TIB115, increased FXa generation levels were observed in TMZ- and LOM-treated cells from all cell lines, in line with previous results. TMZ or LOM treatment in combination with antibody 5G9 prevented FXa generation in all cell lines (*p* < 0.0001 in all LOM-treated cell lines; *p* = 0.0143 in TMZ-treated U-87 cells; *p* = 0.0003 in TMZ-treated U-118 cells), while the amount of generated FXa was reduced by half following chemotherapeutic treatment in combination with Annexin V (*p* < 0.0001 in LOM-treated U-251 and U-118 cells; *p* = 0.0012 in LOM-treated U-87 cells; *p* = 0.0391 in TMZ-treated U-118 cells; Appendix A).

### 3.5. Treatment with TMZ or LOM Induces Expression of Several Proinflammatory Genes in GBM Cells

Finally, the cellular response to chemotherapeutic treatment was investigated since this might contribute to chemotherapy-mediated TF induction. Staining of β-galactosidase activity, a general marker of senescence, was not observed in TMZ- and LOM-treated cells from all cell lines. Nevertheless, TMZ and LOM treatment resulted in significantly increased expression levels of the proinflammatory cytokine genes *IL1B* and *CXCL8* in all cell lines tested, as determined by qPCR (Figure 5). Similar results were obtained for *IL6* and *TNFA*, except for *IL6* expression in TMZ-treated U-118 cells and *TNFA* expression in TMZ-treated U-87 cells. This suggests that inflammatory programs, but not senescence, link chemotherapy to increased procoagulant activity in GBM.

## 4. Discussion

GBM patients are at high risk of developing VTE, which is further increased upon exposure to chemotherapy. Several chemotherapeutic agents are known to induce the upregulation of the procoagulant protein TF [9,10]. However, the potential procoagulant effects of the most commonly used chemotherapeutics for GBM—TMZ and LOM—have not been studied to date. Therefore, we determined the effect of TMZ and LOM on TF procoagulant activity and expression in three GBM cell lines, U-251, U-87 and U-118. Additionally, the potential role of the underlying cellular response to chemotherapy was investigated.

Exposure of GBM cell lines to both TMZ and LOM resulted in increased levels of FXa generation in cells, in the supernatant and on EVs (Figure 1), as well as increased TF gene and protein expression (Figure 2). Moreover, a causal link was observed between treatment with TMZ or LOM and TF procoagulant activity using the TF-blocking antibody 5G9, which completely prevented FXa generation by TMZ and LOM treatment in all cell lines (Figure 4). Altogether, this work provides a potential molecular explanation for the increased risk of VTE in GBM patients undergoing chemotherapy through chemotherapy-induced TF procoagulant activity and expression.

The link between chemotherapy and CAT has already been described extensively [8,9,10]. The antineoplastic effect of chemotherapeutic regimens is critical to restrain tumor cell proliferation but also exerts cytotoxicity to nonmalignant cells, leading to adverse off-target effects. Several mechanisms have been proposed, such as disruption of the endothelium and a disturbed balance between anti- and procoagulants [9]. TF is often mentioned as one of the main procoagulant targets that is affected by anti-cancer therapies [10]. Indeed, enhanced TF activity was described for endothelial cells and a variety of cancer cell types upon treatment with several chemotherapeutic modalities, including cisplatin [16,17], doxorubicin [32] and gemcitabine [17,33], as well as increased TF expression levels on leukemia blood cells following treatment with L-asparaginase [34]. However, the effect of TMZ and LOM on TF activity and expression in GBM cells has not been evaluated.

TMZ and LOM belong to the standard of care for GBM. TMZ-treated patients often develop chemoresistance, which is an important reason for GBM treatment failure. This largely depends on DNA repair enzymes such as O6-methylguanine-DNA methyltransferase (MGMT), which removes methyl groups added to DNA purine bases by TMZ [35]. U-251, U-87 and U-118 are TMZ-sensitive GBM cell lines, showing MGMT promoter methylation [36,37]. The half-maximal lethal concentration (LC50), as published in the literature for TMZ-sensitive cell lines, varies largely, ranging from <20 µM to <500 µM for U-251 cells and from 7 µM to <500 µM for U-87 cells [27].

Interestingly, cytotoxicity exerted by LOM, by adding chloroethyl to the O6-position of guanine, also depends on the activity of MGMT [38], suggesting a similar responsiveness of the cell lines tested to both modalities. However, in addition to DNA alkylation, LOM acts as a bifunctional agent by also inducing DNA interstrand crosslinks as well as amino acid carbamoylation, thereby affecting transcriptional, translational and post-transcriptional processes [25]. These non-alkylating mechanisms might not depend on MGMT enzyme activity [39], resulting in higher efficiency and potentially requiring lower concentrations of LOM in comparison with TMZ. Indeed, concentrations used for in vitro studies with LOM-treated GBM cell lines range from 5 to 55 µM [28,29,30].

Here, the concentrations used for TMZ and LOM (100 µM and 30 µM, respectively) are in the mid-range when compared to other studies [27,28,29,30]. The highest effects on FXa generation and TF gene and protein expression in our study are generally exerted by LOM, possibly due to the additional non-alkylating cytotoxicity of nitrosourea compounds. Furthermore, our data indicate that U-251 and U-118 cells are more responsive to chemotherapeutic treatment than U-87 cells. This may be explained by the mutational status of *TP53*. That is, U-251 and U-118 cells harbor *TP53* mutations, whereas U-87 cells do not [40]. Since *TP53*-mutant GBM cell lines are known to be more sensitive to TMZ [41], this could explain why U-87 cells are less sensitive to TMZ treatment when compared to U-251 and U-118 cells. In the same line, *TP53* mutational status may also be a strong determinant for LOM sensitivity.

TF-positive EVs might play an important role in chemotherapy-induced VTE, being released from apoptotic tumor and endothelial cells following cytotoxic chemotherapy and, thereby, exerting a systemic hypercoagulable state. However, data have been confusing. Normal TF-positive EV activity has been reported by Tesselaar et al. in 20 out of 24 cancer patients who presented with VTE during chemotherapy [42]. Furthermore, a recent study in cisplatin-treated patients with metastatic testicular cancer also showed no induction of TF-positive EV activity following chemotherapy [43], while TF-positive EV activity was found to decrease significantly after chemotherapy in 75 out of 122 newly diagnosed patients with multiple myeloma [44]. Similar results were described for two patients with pancreatic cancer and elevated TF-positive EV activity levels prior to chemotherapy [45]. On the other hand, treatment with doxorubicin was found to increase TF-positive EV shedding in breast cancer cell lines MCF-7 and MDA-MB-231 in vitro, with high doses of doxorubicin resulting in a significant rise in thrombogenic activity of isolated EVs [19]. In GBM, we show that both TMZ and LOM increase the procoagulant activity of EVs. The different results reported may reflect differences in chemotherapy, differences in time of isolation after chemotherapy and differences in tumor type. EVs are not tumor-bound and can travel systemically via the circulation to common VTE sites, suggesting that procoagulant EVs might play an important role in chemotherapy-induced VTE in GBM.

TF-mediated procoagulant activity requires association of the TF:Factor VIIa (FVIIa) complex with phospholipids on the outer leaflet of cell membranes. More specifically, the presence of PS markedly increases TF activity [46]. PS normally resides on the inner leaflet of the membrane and is externalized upon cellular responses to, e.g., oxidative stress and infection by flippase activity. Interaction between the TF:FVIIa complex and externalized PS results in TF decryption by promoting a more favorable conformation for FXa generation, thus stimulating TF procoagulant activity [12]. Interestingly, cancer cells often expose high levels of PS on their cell surface due to differential flippase activity [47,48], and the externalization of PS is induced by chemotherapy [49,50]. However, no data are available on the effect of TMZ or LOM on PS externalization in GBM. We show elevated levels of PS, as detected by Annexin V staining, following treatment with either TMZ or LOM in U-118 cells and with only TMZ in the U-251 cell line (Figure 3). Similar results were seen for LOM in U-251 cells. This may have functional consequences as FXa generation was partially obstructed in the presence of Annexin V (Figure 4), suggesting that chemotherapy-mediated TF decryption due to increased levels of externalized PS may contribute to increased TF procoagulant activity in GBM.

The cellular response to chemotherapy might also play a role in chemotherapy-induced TF activity. Therefore, the effect of TMZ and LOM treatment on cellular senescence and inflammation was investigated. Surprisingly, senescence was not found to be induced in our cell lines after treatment with TMZ or LOM. Nevertheless, chemotherapeutic treatment resulted in increased expression levels of the proinflammatory genes *IL1B*, *IL6*, *CXCL8* and *TNFA* (Figure 5), suggesting an inflammatory response to TMZ and LOM in all cell lines tested. This has also been observed for other chemotherapeutic modalities, such as 5-Fluorouracil (5-FU), paclitaxel and cisplatin, which often induce cytokine expression through NFκB [51,52]. Since TF expression and TF-positive EVs are induced in both vascular endothelial cells as well as cancer cells after secretion of inflammatory cytokines [53,54,55], TF induction following treatment with TMZ and LOM in GBM could be an indirect effect of the proinflammatory tumor cell response to chemotherapy.

## 5. Conclusions

In this study, the role of the chemotherapeutic modalities TMZ and LOM in GBM-related VTE was investigated by studying chemotherapy-induced TF procoagulant activity and expression. We propose a causal relation between treatment with TMZ or LOM and TF activity in GBM, which may result in an increased risk of VTE in GBM patients undergoing chemotherapy. By using three different GBM cell lines, we sought to include cellular heterogeneity in our study, since this is an important hallmark of GBM that greatly influences GBM treatment resistance and tumor recurrence. As increased TF procoagulant activity and upregulation were observed following chemotherapeutic treatment in all cell lines tested, we propose that chemotherapy-induced VTE in GBM is controlled by equally increased TF activation. This work may have implications for future GBM treatment since patients with high tumoral TF levels may benefit from thromboprophylaxis to decrease the risk of chemotherapy-related VTE in GBM.

## Figures and Tables

**Figure 1 cancers-15-02347-f001:**
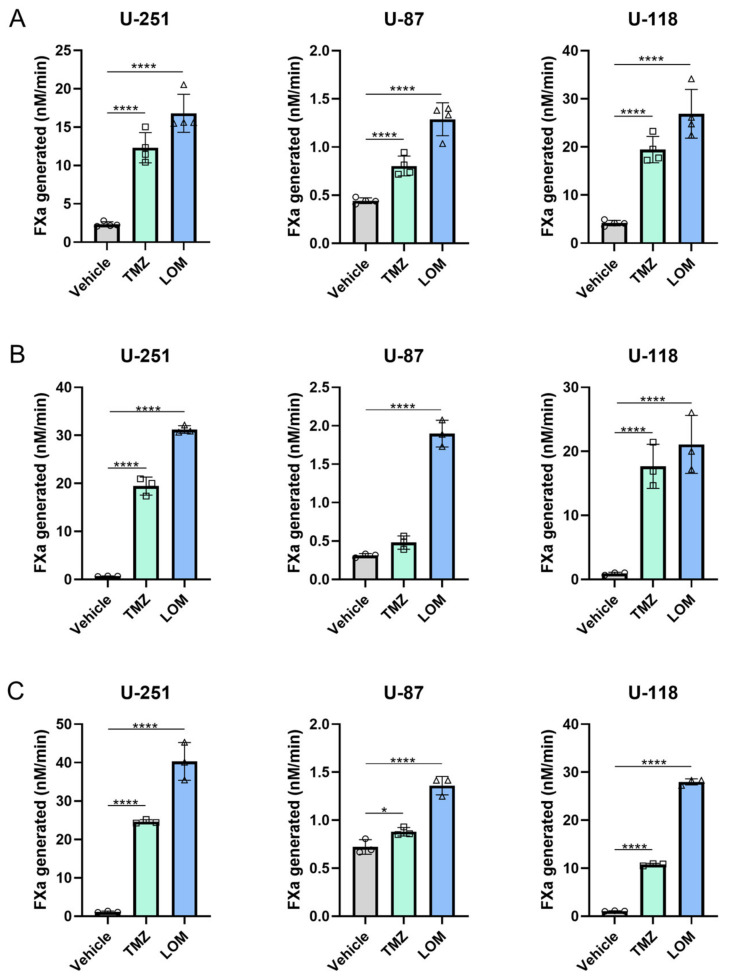
Chemotherapeutic agents TMZ and LOM induce procoagulant activity in GBM cells in vitro. U-251, U-87 and U-118 GBM cells were treated with 100 µM TMZ or 30 µM LOM for 72 h. An FXa generation assay was performed to determine procoagulant activity in substrate-attached cells (**A**), supernatant (**B**) and samples largely enriched for EVs (**C**). The amount of generated FXa was measured 30 min after the start of the reaction in the presence of 1 nM FVIIa and 50 nM FX and corrected for cell number per sample. Representative experiments are shown (*n* = 3) from biological triplicates for each condition. Two-way ANOVA with Tukey’s multiple comparison test was used for statistical evaluation. * *p* < 0.05, **** *p* < 0.0001 vs vehicle (DMSO). Abbreviations: FXa, factor Xa; TMZ, temozolomide; LOM, lomustine.

**Figure 2 cancers-15-02347-f002:**
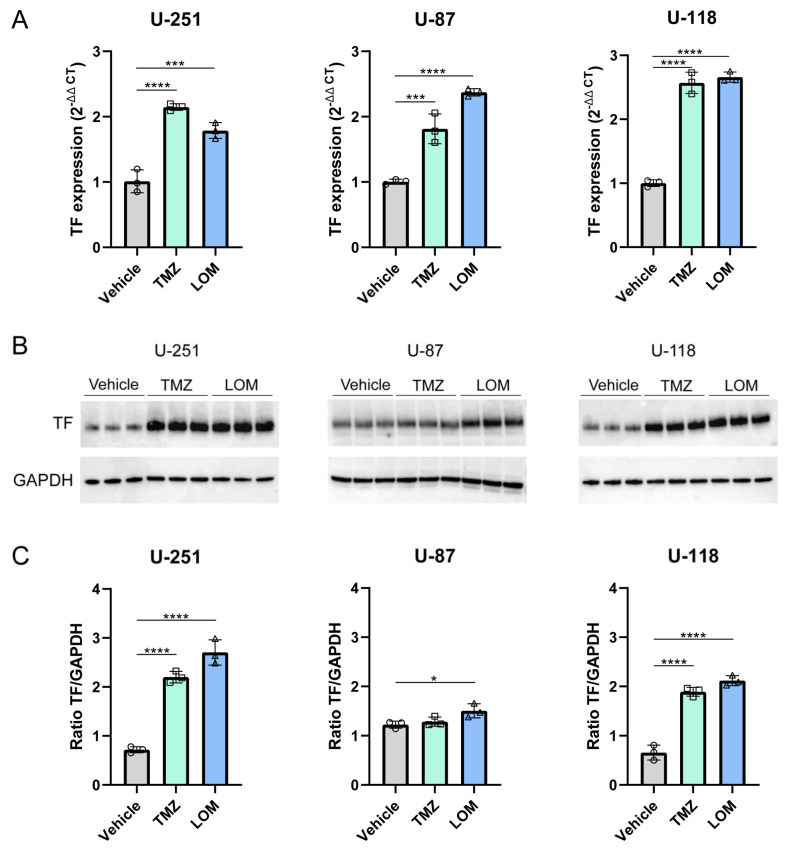
Treatment with TMZ or LOM induces TF expression in GBM cells. (**A**) Treatment with 100 µM TMZ or 30 µM LOM for 72 h resulted in increased TF mRNA expression levels in cells from all cell lines, determined via qPCR. Housekeeping genes *GAPDH* and *ACTB* were used for normalization. One-way ANOVA with Tukey’s multiple comparison test was used for statistical evaluation. *** *p* < 0.001, **** *p* < 0.0001 vs vehicle (DMSO). Representative experiments are shown (*n* = 3) from biological triplicates for each condition in technical triplicate. (**B**) Treatment with 100 µM TMZ or 30 µM LOM for 72 h resulted in increased TF protein expression levels in cells from all cell lines, determined via Western blot. GAPDH was used as loading control. Representative experiments are shown (*n* = 3) from biological triplicates for each condition. (**C**) Relative WB quantification (ratio TF/GAPDH per sample). One-way ANOVA with Tukey’s multiple comparison test was used for statistical evaluation. * *p* < 0.05, **** *p* < 0.0001 vs vehicle (DMSO). Abbreviations: TF, tissue factor; TMZ, temozolomide; LOM, lomustine.

**Figure 3 cancers-15-02347-f003:**
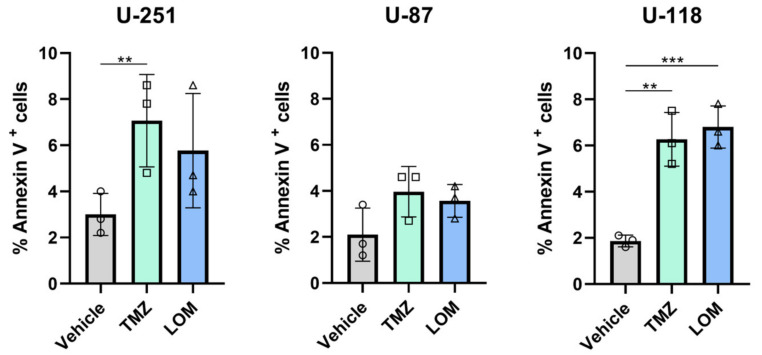
Treatment with TMZ or LOM increases the percentage of Annexin V-positive GBM cells. Cells were incubated with PE-labeled Annexin V for 15 min at RT, followed by several washes with Annexin V binding buffer. The percentage of Annexin V-positive cells was measured using a BD FACSCanto and analyzed using BD FACSDivaTM software. Biological triplicates are shown. ** *p* < 0.01, *** *p* < 0.001 vs vehicle (DMSO). Abbreviations: TMZ, temozolomide; LOM, lomustine.

**Figure 4 cancers-15-02347-f004:**
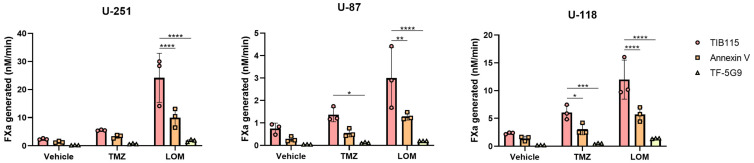
TMZ- and LOM-induced TF activity is fully prevented by TF-blocking antibody mAb 5G9 and reduced by PS-binding Annexin V. FXa generation assay in the presence of TIB115 (IgG1 control), PS-binding unlabeled Annexin V or coagulation blocking TF antibody 5G9. The amount of generated FXa was measured at 30 min after the start of the reaction in the presence of 1 nM FVIIa and 50 nM FX and corrected for the number of cells present per sample. Representative experiments are shown (*n* = 3) from biological triplicates for each condition. Two-way ANOVA with Tukey’s multiple comparison test was used for statistical evaluation. * *p* < 0.05, ** *p* < 0.01, *** *p* < 0.001, **** *p* < 0.0001 vs vehicle (DMSO).

**Figure 5 cancers-15-02347-f005:**
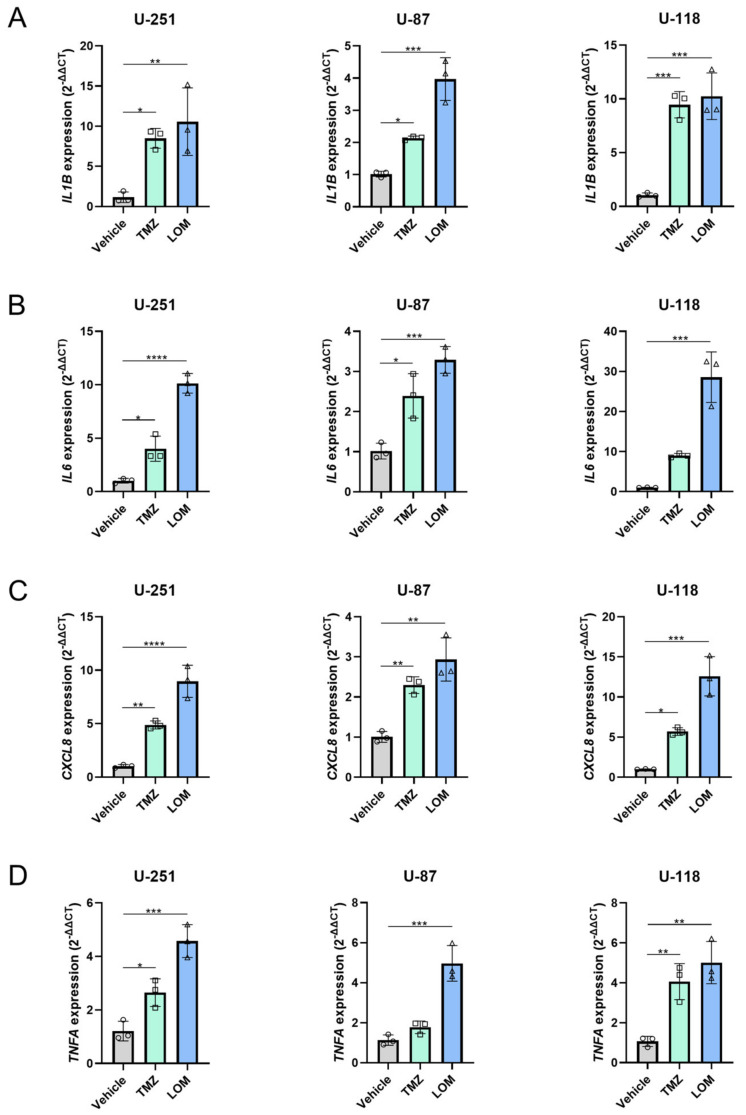
TMZ- and LOM-induced expression of the proinflammatory genes *IL1B*, *IL6*, *CXCL8* and *TNFA* in GBM cells. qPCR analysis showed increased mRNA expression levels of *IL1B* (**A**), *IL6* (**B**), *CXCL8* (**C**) and *TNFA* (**D**) upon treatment of U-251, U-87 and U-118 cells with the chemotherapeutic agents TMZ (100 µM) and LOM (30 µM) for 72 h. Housekeeping genes *GAPDH* and *ACTB* were used for normalization. Representative experiments are shown (*n* = 3) from biological triplicates for each condition in technical triplicate. One-way ANOVA with Tukey’s multiple comparison test was used for statistical evaluation. * *p* < 0.05, ** *p* < 0.01, *** *p* < 0.001, **** *p* < 0.0001 vs vehicle (DMSO). Abbreviations: TMZ, temozolomide; LOM, lomustine.

## Data Availability

The data presented in this study are available in this article.

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
