# Peer review of "Temozolomide and Lomustine Induce Tissue Factor Expression and Procoagulant Activity in Glioblastoma Cells In Vitro"

_cancers, 2023, doi:10.3390/cancers15082347_

Round 1

Reviewer 1 Report

Kapteijn et al. in the manuscript titled “Temozolomide and Lomustine Induce Tissue Factor Expression 2 and Procoagulant Activity in Glioblastoma Cells in Vitro” focused on the effect of temozolomide and lomustine on the gene expression, protein level and the activity of TF in three glioblastoma cell lines. The manuscript is well written but some critical points should be solved and discussed by the authors.

The main doubt is the construction of the experiment and the method of EV isolation.

1.      The authors used different experimental times for the experiments with the cells and the experiments with cell supernatant and EVs.

Line 100: In experiments with supernatant or EVs, medium with FCS was replaced by medium without FCS at 72h after treatment. This was left on the cells for 2h, followed by centrifugation at 1.000g and 4°C for 10 min to remove cell debris and further processing for subsequent read-out assays.

Incubation with FCS-free medium for 2 h might significantly affect the extracellular secretion and concentration/protein level/gene expression profile within the cells. Therefore, additional control should be included in this study to show possible alternations in the cells after incubation with FCS-free medium for 2 h.

2.      The protocol of EVs isolation presented in the manuscript raised doubts. Moreover, the authors did not show any control experiment which could prove that the isolated fraction contained only EVs. Preparation for ultracentrifugation should be preceded by repeated centrifugation at different g values to remove specific fractions of proteins and other cellular elements. Ultracentrifugation only concentrates the EVs fraction, but it is not the main method of EVs isolation. Moreover, the authors did not use EV-free culture medium in their experiments which might affect the final results. The authors should validate their results in further experiments according to the previously published protocols of EVs isolation, i.e. Chhoy P, Brown CW, Amante JJ, Mercurio AM. Protocol for the separation of extracellular vesicles by ultracentrifugation from in vitro cell culture models. STAR Protoc. 2021 Jan 29;2(1):100303. doi: 10.1016/j.xpro.2021.100303. PMID: 33554138; PMCID: PMC7848770.) Results presented in Figure 1 suggested that „EV fraction” was only a concentrated form of cell supernatant.

3.      Why was the 2ΔCT formula used to analyze the gene expression (Figure 2, 5)? The 2 -ΔΔCT formula is used as a standard, which significantly influences the analysis of results. How was normalization applied to 2 housekeeping genes (GAPDH and ACTB)?

4.      Figure 3 and the result section: only statistically significant results should be considered as changes in a number of Annexin V-positive cells/protein levels/gene expression, etc. Using the word "trend" (line 227) or considering the non-statistically significant increase in the experimental group “U-251 LOM” is incorrect.

5.      What are the differences (i.e. genetic mutations, etc.) between glioblastoma cell lines used in this study which might result in the different biological responses to TMZ and LOM. The authors should discuss this issue in the manuscript.

Minor comments: The manuscript contains some editorial errors i.e. double spaces, no spaces between units (1h instead of 1 h), no superscript (1×105 cell), etc.

Reviewer 2 Report

The manuscript with the ID cancers-2308806 entitled "Temozolomide and Lomustine Induce Tissue Factor Expression and Procoagulant Activity in Glioblastoma Cells in Vitro" is an interesting article that points out the effect of two important chemotherapeutic drugs in the treatment of glioblastoma. The authors use glioblastoma cell lines to evaluate the TF-related cellular response. The authors use current literature.

Overall, this is an interesting article on glioblastoma, there are some comments that authors could include:

1- Please include more information on the classification and epidemiology of glioblastoma and its relationship to inflammation. Expanding this information in the introduction section would help to understand the main problems related to glioblastoma.

2- It might be interesting for the authors to describe in more detail the microenvironment within this type of tumor to explain the importance related to TF. 

3- The authors should use an abbreviation for glioblastoma.

4- Why did the authors choose these cell lines, is there another alternative, what happen to glioblastoma stem cells and these drugs?

5- Do the authors use technical replicates?

6- I would also add a graphical abstract.

Round 2

Reviewer 1 Report

The authors addressed all my comments.